# Recent Development in NKT-Based Immunotherapy of Glioblastoma: From Bench to Bedside

**DOI:** 10.3390/ijms23031311

**Published:** 2022-01-24

**Authors:** Yutao Li, Amit Sharma, Jarek Maciaczyk, Ingo G. H. Schmidt-Wolf

**Affiliations:** 1Center for Integrated Oncology (CIO), Department of Integrated Oncology, University Hospital Bonn, 53127 Bonn, Germany; Yutao.Li@ukbonn.de; 2Department of Neurosurgery, University Hospital Bonn, 53127 Bonn, Germany; Amit.Sharma@ukbonn.de (A.S.); Jaroslaw.Maciaczyk@ukbonn.de (J.M.); 3Department of Surgical Sciences, University of Otago, Dunedin 9054, New Zealand

**Keywords:** glioblastoma, immunotherapy, invariant NKT, cytokine-induced killer cells, blood–brain barrier, blood–brain tumor barrier, tumor infiltration lymphocytes, overall survival

## Abstract

Glioblastoma multiforme (GBM) is an aggressive and dismal disease with a median overall survival of around 15 months and a 5-year survival rate of 7.2%. Owing to genetic mutations, drug resistance, disruption to the blood–brain barrier (BBB)/blood–brain tumor barrier (BBTB), and the complexity of the immunosuppressive environment, the therapeutic approaches to GBM represent still major challenges. Conventional therapies, including surgery, radiotherapy, and standard chemotherapy with temozolomide, have not resulted in satisfactory improvements in the overall survival of GBM patients. Among cancer immunotherapeutic approaches, we propose that adjuvant NKT immunotherapy with invariant NKT (iNKT) and cytokine-induced killer (CIK) cells may improve the clinical scenario of this devastating disease. Considering this, herein, we discuss the current strategies of NKT therapy for GBM based primarily on in vitro/in vivo experiments, clinical trials, and the combinatorial approaches with future therapeutic potential.

## 1. Introduction

Glioblastoma multiforme (GBM), the most common malignant primary brain tumor, is highly diffusive and infiltrative in nature. Classified as World Health Organization grade IV astrocytoma [1], it remains incurable to date due to the low efficacy of standard therapies (surgical resection, radiotherapy, and chemotherapy with temozolomide). The pronounced recurrence rate, unfavorable prognoses, and post-treatment symptoms pose a serious clinical challenge. Of note, patients usually develop GBM sporadically, and any potential link to inherited genetic variations that may increase the risk for primary adult glioma has also been discussed [2]. Furthermore, no risk factor responsible for a large proportion of GBM cases has been identified either.

GBM ranks high on the cancer spectrum owing to epigenetic-based prognostic markers, e.g., isocitrate dehydrogenase 1 (IDH1) gene mutations with high global DNA methylation levels and hypermethylation in the O6-methylguanine DNA methyltransferase (MGMT) gene promoter region, which play a pivotal role in the making of clinical decisions. Moreover, the routine analysis of the methylation status of the promoter region of the MGMT gene has been shown to be beneficial for an effective response to temozolomide-based chemotherapy [3]. Interestingly, several microRNAs (miRNAs) have been identified in GBM and are thought to play important roles in initiation, progression, and response to therapy [4,5,6]. Of interest, these miRNAs are often combined with MGMT status to assign patients to high and low-risk groups [7,8]. In addition, glioblastoma-associated X-linked tumor suppressor genes (FLNA and FOXP3) have been shown to spatially interact with specific autosomal loci (DRD2 and RORC) [9]. Along with inter- and intratumoral heterogeneity, both cellular and molecular heterogeneity has gained attention in GBM. In fact, glioblastoma stem cells (GSCs), a small cell subpopulation in GBM, are held responsible to some degree for tumor heterogeneity, treatment resistance, and recurrence. Koch et al. recently assessed the therapeutic potential of glutaminase (GLS) inhibitors on GSCs in vitro and validated them as a target with low adverse effects [10]. Among other advances, it has been shown that the blocking C-repeat/DRE binding factor 1 (CBF1) in glioblastoma cells can lead to the efficient suppression of epithelial-to-mesenchymal transition (EMT) activators, including zinc finger E-box-binding homeobox 1 (ZEB1) [11]. Pharmacological inhibition of the Wnt pathway by porcupine inhibition also showed increased susceptibility to temozolomide (TMZ, the first-line chemotherapeutic agent that has been used to treat gliomas for more than a decade) treatment, presumably involving downregulation of aldehyde dehydrogenase 3 family member A1 (ALDH3A1) [12]. More recently, molecular monitoring of GBM immunogenicity using a combination of Raman spectroscopy and chemometrics was shown to track T cells and monocytes [13]. While preclinical models have advanced our knowledge for ranking GBM in the spectrum of the cancer landscape, they have shown less efficacy on the treatment side.

Another layer of complexity in GBM relates to microanatomical compartments representing specialized tumor niches within the tumor microenvironment that regulate metabolic needs, immune surveillance, survival, and invasion, as well as maintenance of cancer stem cells [14]. Certainly, the contribution of complex microenvironments surrounding glioblastoma tumors, which are primarily composed of cell-produced soluble factors, extracellular matrix components, resident cells (e.g., astrocytes, endothelial cells, pericytes, and microglia) and recruited cells (e.g., bone marrow-derived macrophages and tumor-infiltrating lymphocytes (TILs)) cannot be excluded [15]. It should be mentioned that there has been considerable progress concerning therapeutic targeting of the microenvironment in GBM [16]; however, clinical implementation is pending.

Over the years, several clinical trials have been conducted on this disease, including immunotherapeutic approaches that are intended to promote the antitumor immune response in patients. However, they did not yield the desired outcome. Herein, we will decipher the function of NKT cells in GBM and review the prospective of cytokine-induced killer (CIK) cells in GBM, with an emphasis on the role of NKT cells and recent NKT-based innate and adaptive immune therapy in GBM.

## 2. Landscapes of Unique Immune Suppression in Glioblastoma

### 2.1. The Blood–Brain Barrier and the Blood–Brain Tumor Barrier (BBTB)

The impaired blood–brain barrier (BBB) is a physiological obstruction to the delivery of drugs to the brain parenchyma and central nervous system (CNS) in the GBM, which might hinder the accessibility of chemotherapeutics to tumor cells [17]. The BBB is a complex interaction between endothelial cells, astrocytes, pericytes, basal lamina, and extracellular matrices (ECMs) [18,19], which precisely regulates the movement of ions, molecules, drugs, and cells between the brain and blood vessels (Figure 1). However, it is reasonable to assume that this precise hemostasis regulation may be impaired in GBM. Notably, some factors released from GBM disrupt the BBB barrier and are considered to be ‘immune privileged’ [20]. For instance, Sema3A in extracellular vesicles (EVs) released by patient-derived glioblastoma cells disrupt the endothelial barrier [21]. Radiation has also been the cause of BBB disruption [22], and some chemotherapy agents, such as etoposide and cisplatin, were found to be in high levels in tumors compared to neighboring tissues. Moreover, TMZ is known to increase the BBB permeability of drugs that are normally effluxed by Pgp (P-glycoprotein) back into the bloodstream [23]. The authors demonstrated that TMZ (at therapeutic concentration) increased the transport of Pgp substrates across human brain microvascular endothelial cells and decreased the expression of Pgp.

The scenario is much more complex for the blood–brain tumor barrier (BBTB), which reflects the complex regional heterogeneity of immune cell populations compared to the BBB [33]. The BBTB is generally considered ‘leakier’ than the BBB, which is characterized by aberrant pericyte distribution and loss of astrocytic endfeet and neuronal connections [34,35]. It has been confirmed that BBB integrity is disrupted by invading glioma cells [17,36]. In addition, the barrier-like structure of the BBTB differs from the BBB primarily with respect to the formation of brain tumor capillaries and is a major obstacle to successful drug delivery. The disruption of the BBB and the visual heterogeneity in GBM tissues can be observed by contrast-enhanced MRI. A study based on similar assumptions showed that there was minor uptake of contrast media in necrotic tumor areas, whereas substantial microvascular leakage, in contrast, was observed in the tumor interstitial space [37]. Currently, fluorescence-based imaging of brain tumors is emerging as an option for the visualization of the BBB and BBTB, especially as a potential optical surgical tool [38].

### 2.2. Molecular Heterogeneity

The concept of ‘tumor heterogeneity’ usually encompasses both inter-tumor and intra-tumor heterogeneity [39]. In addition, cancer-related (mutated) genes are tightly linked and can reshape the genome in different types of cancer [40]. With regard to the Cancer Genome Atlas Consortium (TCGA), molecular classification of nearly 600 GBM tumors, mutated genes TP53, epidermal growth factor receptor (EGFR), IDH1, and phosphatase and tensin homologue (PTEN) [41], as well as the three core pathways, namely p53, RB, and receptor tyrosine kinase (RTK)/Ras/phosphoinositide 3-kinase (PI3K) signaling, were reported [42]. Furthermore, Wang et al. defined three tumor-intrinsic transcriptional subtypes designated as proneural (PN), mesenchymal (MES), and classical (CL) within the same IDH wild-type glioblastoma, which are partly shaped by the tumor-associated immuno-environment [43]. Consistent with these observations, Neftel et al. used a full-length scRNA-seq (SMART-Seq2) approach to show that each tumor contains multiple cellular states, named, accordingly, astrocyte (AC)-like, oligodendrocytic precursor cell (OPC)-like, neural progenitor cell (NPC)-like and mesenchymal (MES)-like [44]. TCGA-CL and TCGA-MES subtypes correspond to tumors enriched for the AC-like and MES-like states, respectively; the TCGA-PN subtype corresponds to the combination of two distinct OPC-like and NPC-like ‘hybrid’ cellular states. Some of these genetic alterations can promote specific cellular states, for instance, EGFR drives an AC-like program and CDK4 controls an NPC-like program in mouse neural cells. Therefore, the landscape of GBM heterogeneity appears to be more complex than previously thought. Beyond that, assessment of tumor heterogeneity at a single time point (spatial heterogeneity) and/or along the clinical recurrence/evolution of GBM (longitudinal heterogeneity) may contribute to further improvements in individualization of therapy [45]. Taken together, assessment of heterogeneity in GBM may decipher immune suppression and strongly undermine the efficacy of any ongoing/scheduled therapy for the individual patients.

### 2.3. Glioblastoma Tumor Microenvironment

In GBM, a paucity of tumor-infiltrating lymphocytes (TIL) has long been inferred because of the sequestration of T cells in bone marrow due to a loss of surface spingosine-1-phosphate receptor 1 (S1P1) [24]. On the other hand, intracranial glioma disrupts thymic homeostasis, resulting in an imbalance of double-positive and CD4^+^ and CD8^+^ single-positive T cell subsets and induces T cell apoptosis by induction of Notch-1/ Jagged-1 pathway in vivo [46]. In particular, GBM elicits severe T cell exhaustion among CD8^+^ TILs with prominent upregulation of inhibitory immune checkpoints PD-1, LAG-3, TIGIT, and CD39, and the function of TILs from murine GBM, such as the production of IFN-γ and IL-2, was impaired [47]. However, recent research demonstrated that increased markers of memory/antigen experience (CD45RA, CD27, and CD127) in the peripheral blood of PD-1^+^ T cells are found in glioblastoma patients, suggesting that the PD-1^+^ peripheral T cells of GBM patients exhibit activation functions compared to GBM TILs [25].

Natural killer group 2 member D (NKG2D) is known to induce cytotoxicity and cytokine production by NKT cells upon binding to its ligands. It is also well established that NKG2D recognizes a number of legends (MHC class I polypeptide-related sequence A (MICA), MHC class I polypeptide-related sequence B (MICB), and UL16-binding proteins 1–6 (ULBP1-6)). Of interest, these ligands are also expressed on human glioma cells, in vitro [26], in vivo [27], and on glioma stem cells [15,48]. Recently, one study not only confirmed the high expression of NKG2DL in human glioma cell lines, cancer stem cells, and tumor samples, but also suggested that CAR-T cells expressing NKG2D are an encouraging therapeutic approach for glioma patients [49]. An interesting study described the potential mechanism of immune escape, in which glioblastoma cells produce a soluble protein LDH5 that induces the expression of NKG2D ligands on the surface of healthy myeloid cells [50]. Of note, combining NKG2D-based immunotherapies with TMZ or irradiation (IR) has also been suggested [51].

Given that TME in GBM provides an unfavorable niche for the function of NK cells, understanding the orchestration of TME constituents is also crucial. In particular, it is important to mention the interaction of microglia and astrocytes (glioma-associated microglia/macrophages, GAMs) in GBM [52,53], as it has already been shown that targeting these cells can have a substantial impact on GBM. One study suggested that the immunomodulatory properties of NK cells, mainly their ability to secrete pro-inflammatory cytokines and to influence microglial and macrophage activity, could be used to enhance the efficacy of passive immunotherapy targeting tumor-associated antigens in GBM [54]. Similarly, one study reported that GAMs, stimulated by the brain-derived neurotrophic factor (BDNF), increased NK cell infiltration and activation, thus contributing to the modulation of glioma expansion [55]. An interesting study showed that phytosomal curcumin (CCP) treatment caused a GBM tumor to acquire M1-type macrophages and activated NK cells [56]. The authors further suggested that M1 microglia-derived MCP-1 may recruit both cell types (M1 macrophages and activated NK cells) into the GBM tumor.

In addition, myeloid-derived suppressor cells (MDSC) and glioblastoma stem-like cells (GSCs) can hinder the immune response of T lymphocytes [28]. Specifically, GSCs perform complex crosstalk in their perivascular and perinecrotic niches to maintain their properties and stimulate cellular plasticity, especially to express CD133 in the hypoxia niches, where stabilization of hypoxia-inducible factor-1 (HIF1) α is important and consequently appears to contribute to tumor survival and progression [29,57]. Some, such as neurotensin, growth differentiation factor-15 (GDF-15), sphingosine-1-phosphate (S1P), and infection with cytomegalovirus have a direct influence on the GBM-related tumor microenvironment [58]. The possible immune cell mediators potentially suppressing the microenvironment in GBM are presented in Figure 1.

## 3. Natural Killer T (NKT) Cells

It is well established that NKT cells are a subset of T cells that co-express the αβ T cell receptor but also a variety of molecular markers that are typically associated with NK cells, such as NK1.1. The term ‘natural killer T (NKT) cells’ was first published in 1995 and broadly defined murine T cells with the presence of natural killer (NK) cells marker NK1.1. [59]. Based on their TCR repertoire, Dale I. Godfrey et al., classified NKT cells into three subsets: type I cells (classical NKT cells), type II cells (non-classical NKT cells), and NKT-like cells (CD1d-independent NK1.1^+^ T cells) in mice [60]. These three distinct subpopulations, expressing different NK-associated receptors, are summarized in Table 1.

The most extensively studied are the type I NKT cells, which express an invariant TCR α-chain (Vα14-J18 in mice, Vα24-Jα18 in humans) and recognize glycolipid α-galactosylceramide (α-GalCer) presented by the non-polymorphic MHC class I-like molecule, CD1d, as described in Table 1. As mentioned above also, the type I NKT cells can also be activated by a strong agonist, such as α-galactosylceramide (αGalCer), resulting in rapid cytokine release of IL-4, IL-13, and IFN-γ. In contrast to type I NKT cells, knowledge about type II NKT cells is limited. Type II NKT cells are also CD1d-restricted, but they are distinguished from type I NKT cells because they express neither the invariant TCR α-chain that characterizes type I NKT cells, nor do they recognize α-GalCer. A population of type II NKT cells that recognize and respond to the microbial antigen α-glucuronosyl-diacylglycerol (α-GlcADAG) presented by CD1d was described by Catarina F. Almeida et al. [96]. Type II NKT cells express a more diverse set of TCR α chains and recognize more diverse antigens, such as sulphatide [67], β-glucosylceramide (β-GlcCer), phosphatidylglycerol (PG), diphosphatidylglycerol (DPG), lysophosphatidylcholine (LPC), and lysophosphatidylethanolamine (LPE). The function of type II NKT cells is poorly understood, although some evidence suggests that they play an immunosuppressive role in some studies [92]. For instance, NKT II cells produce IL-13 in response to tumor growth, resulting in the excretion of TGF-beta from myeloid cells that inhibits cytotoxic T cell-mediated tumor immunosurveillance in several mouse tumor models [93]. However, novel CD4^+^ and DN type II NKT cells that express NKG2 receptors have been demonstrated to produce the T_H_1-like cytokine IFN-γ, suggesting that this novel subset of CD4^+^ and DN type II NKT cells are biased toward the typical NKT I cells and have Th1-like cytokines production [91]. Moreover, targeting intestinal type II NKT cells using orally delivered sulfatide facilitates intestinal immunoglobulin A (IgA), T helper 1 (Th1), and T helper 17 (Th17) responses in mice [94].

NKT-like cells are a subset of αβ T cells that express NK-associated receptors, which exhibit a highly specialized effector memory phenotype [95]. The functional receptors of NKT-like cells form a complex repertoire of activatory (NKG2C, NKG2D, NKp30, NKp44, and NKp46) and inhibitory (CD158a, CD158b, KIR3DL1, and NKG2A) receptors, which recognize ligands on the surface of target cells [79,80]. Upon activation, according to the expression of CD4 and CD8, NKT cells display distinct Th1 and Th2 cytokine profiles. CD4^+^ NKT cells produce Th1 and Th2 cytokines, and CD4^−^ NKT cells that include double negative (DN, CD4^−^CD8^−^) and CD8^+^ NKT cells primarily produce Th1 cytokines, such as IFN-γ. Furthermore, NKT-like cells mediate non-MHC-restricted target cell lysis by exocytosis of perforin and granzyme [79,97] or through polarized degranulation, which controls the delivery of FasL to the cell surface and finally regulates FasL-mediated apoptosis [98].

## 4. NKT Cells in Glioblastoma

Type I NKT cells play a pivotal role in anti-tumor immunity. After expansion with IL-2 and α-GalCer (KRN7000, a synthetic glycosphingolipid originally isolated from a marine sponge) from PBMC of healthy donors, type I NKT cell-mediated cytotoxicity was induced by both CD1d-positive glioblastoma cell lines or CD1d-positive patient-derived glioblastoma cells in vitro, with significant increases in the production of IFN-γ, TNF-α, granzyme B, and IL-4 [99]. Although in this investigation the authors reported that 10 out of 15 patients expressed CD1d in glioblastoma cells, infiltration of Vα24+ type I NKT cells was not detected in any patient whereas infiltration of CD3^+^ T cells into brain tumor tissue was observed in 14 patients. In comparison to typical T-cell receptor–peptide antigen–MHC complexes, type I NKT TCRs adopted parallel docking modes, positioned over the extreme end, directly above the F′ pocket of the CD1d-antigen binding cleft to form a lock-and-key NKT TCR–CD1d–α-GalCer complex [100]. As the tissue with the second-highest lipids content, the brain mainly uptakes phosphatidylethanolamines, which are the most abundant phospholipids, followed by phosphatidylcholine, phosphatidylserine and phosphoinositides in correlation to either a structural component of mitochondrial membranes or signal transduction. Based on this specific physiological characteristic in the brain, GBM mainly exhibited enrichment of glycosphingolipid metabolic progress in comparison to the enrichment of phosphatidylinositol metabolic progress in lower-grade gliomas [101]. Together, even though the extent of type I NKT infiltration in glioblastoma lesions was undetectable [99], the potential intracranial introduction of type I NKT cells are required against CD1d-expressing glioblastomas in tumor microenvironments abundant in endogenous glycosphingolipids.

In addition to some barriers mounting an effective immune response against GBM, one study performed T cell mRNA expression profiles in GBM patients and found that genes associated with T cell activation were significantly reduced in CD4^+^ and CD8^+^ T cells, while expression of inhibitory genes was increased in the immunosuppressive Treg subset [102]. Similarly, glioma-derived miR-92a induces IL-6^+^ IL-10^+^ NKT cells, which exhibit the suppressive function of cytotoxic CD8^+^ T cells [103]. Consistent with these results, Allen Waziri et al. demonstrated that a significant proportion of tumor-infiltrating lymphocytes (TIL) within GBM were CD4 single-positive CD3^+^ CD56^+^ T cells producing IL-4, IL-13, which suggests that suppression of CTLs may be regulated within the GBM microenvironment via inducing GM-CSF secretion by myeloid suppressor cells. CD56^+^ T cells identified within GBM were not type I NKT cells, as they demonstrated diverse TCR expression. Conversely, CD4^+^ CD25^high^ “T_regs_” demonstrate only a modest proportional increase within GBM compared to the PBMCs of glioblastoma patients. This evidence elucidates the capacity of GBM recruitment and activation of CD4^+^ CD56^+^ NKT cells is unique in comparison to the vast majority of single CD8-positive CD3^+^ CD56^+^ cells in the PBMCs of patients with GBM [32].

Recently, it has been demonstrated that macrophages, NK and NK T cells, MDSCs, and Tregs were correlated with poorer glioblastoma patient prognoses in contrast to the beneficial role of CD8^+^ T cells in a meta-study [104]. One of the potential reasons for the low iNKT cell numbers and the endogenous immune response imbalance in the growth of glioblastomas might be their inefficient homing into malignant tissues [105] or thymus homeostasis disruption [24], which may be overcome by autologous or allogeneic iNKT or CIK transplantation as an immunotherapeutic approach.

## 5. Preclinical NKT-Mediated Immune Therapy in Glioblastoma

Type I NKT cell-targeted adaptive therapies, such as autologous α-GalCer-pulsed antigen-presenting cells, are promising options for cancer treatment. An irradiated GL261 murine glioma loaded with α-GalCer was implanted intravenously to activate iNKT cells found predominantly in the spleen and liver. The a-GalCer-loaded whole tumor vaccine primes iNKT in the lung-draining lymph nodes with the release of cytokines, including IL-4, IL-13, and IFN-γ, into the serum and is effective against established intracranial tumors but requires depletion of regulatory T cells (Treg). This vaccine elicits a CD4^+^ T-cell–mediated immune response and long-term survival [106] (Figure 2A).

In a CD1d-positive U251, orthotopic xenogenic model of glioblastoma, intracranially co-injected human type I NKT cells with α-GalCer significantly prolonged the survival of tumor-bearing mice compared with α-GalCer alone. In addition, type I NKT cells injected with or without α-GalCer tended to delay tumor growth compared with the control injection. In contrast, type I NKT cells failed to hinder tumor growth of CD1d-negative U87 cells in the intracranial injection model, suggesting that human type I NKT cells exert direct cytotoxicity against CD1d-expressing glioblastoma cells [99]. This study indicates that α-GalCer and iNKT cell-based cancer immunotherapy has anti-glioblastoma therapeutic potential (Figure 2B).

While preclinical mouse models for α-GalCer-dependent iNKT cell-based cancer immunotherapy led to anti-tumor responses in GBM [106], the anti-tumor properties were CD1d-restricted in vitro [99]. It has been shown that the expression level of CD1d on stem-like cells derived from patient glioblastoma was lower than that on the original patient glioblastoma cells. However, all-trans retinoic acid (RA) can induce CD1d expression in glioblastoma stem-like cells, which promotes iNKT cells to exhibit higher cytotoxicity against α-GalCer (alpha-galactosylceramide)-pulsed patient glioblastoma stem-like cells [100]. Therefore, α-GalCer-dependent iNKT cell-based cancer immunotherapy targeting high CD1 expression GBM is a promising therapeutic strategy for the future. Further study of α-GalCer analogs and humanization of the CD1d/iNKT cell murine model [108] might optimize this innate immunotherapy. Another potential novel strategy for GBM might be expanded NKT cells by autologous mature dendritic cells (DCs) loaded with α-GalCer [30]. There is some evidence demonstrating that DCs loaded with α-GalCer are effective vaccines against B16 murine melanoma [31] and phase I clinical immunogenic melanoma patients [109]; therefore, either α-GalCer or complexation with tumor antigen-loaded DC-CIK cells might provide a good platform to fulfill this perspective based on the feasibility and encouraging efficacy in previous CIK clinical trials.

To overcome the disadvantages of suppressive tumor environments, a PD-1/PD-L1 blockade might be a valuable option to enhance the cytotoxicity of iNKT cells. In this context, a study showed that the co-administration of anti-PDL1 antibody and alpha-galactosylceramide (αGalCer)-pulsed APCs enhances iNKT cell-mediated antitumor immunity [110]. Similarly, a critical role for the PD-1/PD-L costimulatory pathway in the alpha GalCer-mediated induction of iNKT cell anergy as a possible target for immunotherapies has been discussed [111]. Since PD-L1 expression is a predictive biomarker for CIK cell-based immunotherapy and the PD-1 blockade has been shown to enhance CIK cell cytotoxicity [112], the PD-Ls/NKT/CIK axis needs consideration. In order to delete the effect of Tregs on anti-tumor immunity, pre-administration of depleting anti-CD25 monoclonal antibodies prior to α-GalCer vaccination increased α-GalCer-induced prophylactic anti-tumor function in the GL261 murine glioma model, as mentioned above [106]. More studies are compulsory to delineate the relationship between iNKT/NKT cells and Tregs in order to manipulate iNKT/CIK-mediated cytotoxicity in glioblastomas. On the other hand, chimeric antigen receptor (CAR) T cells are part of an ongoing novel strategy for treating patients with glioblastomas. There are five clinical phase I/II trials investigating IL-13Rα2-, EGFRvIII-, and HER2-directed CAR T cells for the treatment of glioblastomas [113]. However, the anti-tumor efficacy is not yet satisfactory. With the increasing understanding of CAR-CIK cells technology, a novel therapeutic approach against GBM needs to be thoroughly investigated.

## 6. CIK Cell Adaptive Immunotherapy

### 6.1. Characteristics of CIK Cells

Cytokine-induced killer (CIK) cells were first described in 1991 by Schmidt-Wolf et al. [114]. The same group also performed the first clinical trial using these cells in lymphomas [115,116]. CIK cells exhibit similarities to classical invariant iNKT cells with deficiencies in 2B4 stimulation and in the costimulation of CD3 with NKG2D. The same group recently showed that NKG2D engagement alone is sufficient to activate CIK cells, while 2B4 only provides limited coactivation [117]. To date, more than 80 clinical trials with CIK cells have been conducted, ranging from solid tumors to blood malignancies (clinicaltrial.gov). CIK cells are primarily heterogeneous cells derived from PBMCs of healthy donors or patients; after sequential incubation of IFN-γ and IL-1β IL-2, CIK cells are expanded in vitro for 14 days, with dual functional effector T cells and NK-like cells [114]. Notably, CD3^+^ CD56^+^ CIK cells are terminally differentiated CD8^+^ effector memory T cells, derived from proliferating CD3^+^ CD56^−^ CD8^+^ T cells. They express polyclonal T cell receptor Vβ chains, with a high level of NKG2D and low levels of NKp44 (18%) and NKp30 (10%) [118]. Chieregato et al. reported that a CD56^+^ cell fraction after immunomagnetic selection is composed of NK^bright^ cells (CD3^−^CD56^+bright^) and two subsets of NK-like T cells (CD3^+^ CD56^+^), called NK-like T CD56^dim^ and NK-like T CD56^bright^. The cytotoxic capability was mainly exhibited by the NK^bright^ subpopulation and inversely correlated with NK-like T CD56^dim^ cells in vitro [119].

CD3^+^ CD56^+^ CIK cells are considered not to be type I NKT cells based on the outcomes of only 4% Vα24 surface expression on CD3^+^CD56^+^ T cells after exposure to glycosphingolipid KRN7000 [120], which was described by Gütgemann et al. However, there is no single unique molecule or set of molecules to define the phenotype of type II NKT cells. In addition, sulphides/CD1d-tetramers are not available due to their instability [121], which also makes it difficult for CIK cells to be classified as type II NKT cells. Taken together, it has been unclear whether these different NKT cell subsets of CIK cells possess distinct cytokine profiles and functional capabilities in vitro or vivo. It will be vital to determine the precise correlation between the phenotype and the function of subsets of CIK cells before therapeutic strategies are pursued.

### 6.2. CIK In Vitro and In Vivo Experiments in Glioblastoma

The first in vitro experiments with CIK cells on cytotoxicity against two pediatric glioblastoma multiforme cultured cell lines (G74 and G77) were reported in 2003. Median lytic activity rates of CD3^+^CD56^+^ cells against G74 and G77 measured by LDH release cytotoxic assays were 62.5–64.5% compared to normal peripheral mononuclear cells, which were only 8.5–10%, respectively [122]. The first CIK preclinical research on glioblastoma was recorded in 2011 [123]. hCIK cells (1 × 10^5^, 1 × 10^6^, or 1 × 10^7^, once a week for four weeks) were injected into the tail veins of immune-compromised mice bearing U-87MG tumors in their brains and reduced tumor growth significantly, by 44%, 54% and 72%. Moreover, hCIK cell (1 × 10^7^ once a week for four weeks) and TMZ (2.5 mg/kg, daily for 5 days) combination treatments further increased tumor cell apoptosis and decreased tumor cell proliferation and vessel density (*p* < 0.05), creating a more potent therapeutic effect (95% reduction in tumor volume) compared with either hCIK cells or TMZ single therapy (72% for both, *p* < 0.05) (Figure 2C). In 2015, Ma et al. investigated the efficacy of CIK cells armed with the bispecific antibody anti-CD3 x anti-EGFR (EGFRBi-Ab) to target EGFR-positive glioblastoma in vitro and vivo. EGFRBi-armed CIK cells secreted significantly higher levels of IFN-ϒ, TNF-α, and IL-2 compared to their unarmed CIK cells. Furthermore, in glioblastoma xenograft mice, an infusion of 5 × 10^7^ EGFRBi-armed CIK cells per mouse successfully inhibited the growth of glioblastoma tumors [107] (Figure 2C).

### 6.3. CIK Clinical Trials in Glioblastoma

There are four registered CIK clinical trials in GBM in the ClinicalTrials.gov database of the NIH: a phase I/II clinical trial evaluating DC-CIK treatment of malignant gliomas following tumor resection and radiotherapy (NCT01235845, listed in 2010, not yet recruiting); a study of CIK in combination with temozolomide with and without radiation in adults with advanced malignant gliomas (NCT02496988, listed in 2015, not yet recruiting); a study of CIK in combination with temozolomide with and without radiation in adults with stage I-II gliomas (NCT02494804, listed in 2015, not yet recruiting). Unfortunately, all of these studies are not yet recruiting.

In a randomized, open-label, multicenter phase III trial (NCT00807027, started in 2008, finished in 2012), the researchers evaluated the efficacy and safety of adoptive immunotherapy with autologous CIK cells given with standard TMZ treatment in patients in Korea with newly diagnosed glioblastomas [124]. In this study, 91 glioblastoma patients were randomized to the CIK immunotherapy group and 89 patients were randomized to the control group. Patients in the CIK immunotherapy group received the CIK cell adaptive immune therapy containing 6.55 × 10^9^ cells per cycle combined with standard TMZ chemoradiotherapy. Conversely, the patients in the control group were treated with standard TMZ radio-chemotherapy. The clinical trial has been depicted in Figure 3.

In the intention-to-treat analysis, median PFS (progression-free survival) was improved by 8.1 months in the CIK immunotherapy group compared to the control group with 5.4 months. Additionally, Grade 3 or higher adverse events did not show a significant difference between groups. However, the CIK immunotherapy group did not show evidence of a beneficial effect on overall survival. This study indicates the feasibility and safety in phase III CIK cells-based trials with promising efficacy in glioblastoma patients. However, the limited size of the study population and the lack of an investigation into the molecular background of glioblastomas means that additional observations are needed. Recently, it has been suggested that optimization of CIK cell therapy in combination with other contemporary cancer therapies in a complementary manner (rather than in competition) may help to combat cancer [125].

## 7. Conclusions and Future Perspectives

This review highlights the role of NKT cells in gliomas, with particular emphasis on their subtypes. Based on the efficacy of NKT cells in previous studies, it is reasonable to speculate that there is an urgent need to enhance their function, primarily by regulating their activation throughout tumor progression, in order to maintain their anti-tumor functions. This, in turn, may help to balance the activity of NKT cells within the intracellular signaling cascade, thereby reducing their chance of immune escape. Additionally, we need a better understanding of the orchestration of TME components, as they appear to be the main culprits in providing an unfavorable niche for the function of immune cells, including NKT cells. Considering the biological similarities between NKT and CIK cells, a combinatorial approach can be considered to enhance immune surveillance. In this respect, the simultaneous expansion and possible activation of these NKT/CIK cells can be established and tested in preclinical models. The preliminary results from the combination of iNKT-targeted dendritic cell (DC) vaccines appear to be encouraging in glioblastoma treatment, and further consideration of α-GalCer-loaded DC-CIK cells may also help to widen this avenue. Clearly, more research is required to elucidate the therapeutic potential of NKT-based approaches. Crucially, the extent to which patient genetics contribute to variable responses to therapies based on these cells also needs to be reconsidered.

## Figures and Tables

**Figure 1 ijms-23-01311-f001:**
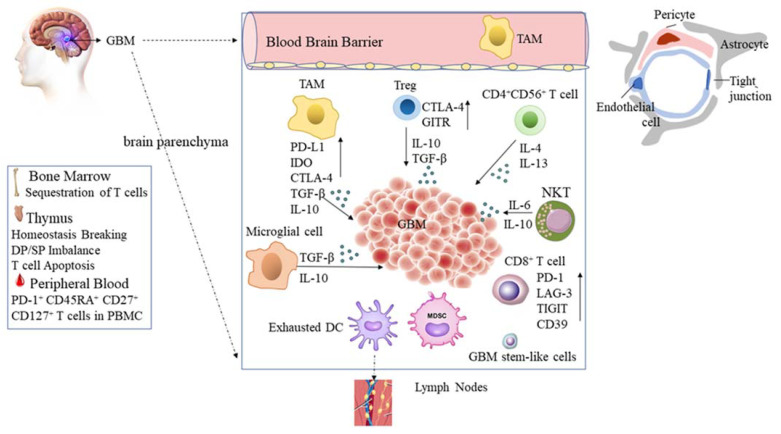
Immunosuppression in glioblastoma. The blood–brain barrier is formed by vascular endothelial cells, pericytes, and astrocytes. In the systemic human body, the sequestration of T cells in the bone marrow, the hemostasis breakdown in double negative (DP)/single positive (SP) T cells in the thymus contribute to the immune-suppressive environment. In the local brain parenchyma, the glioblastoma exits four morphic cell types, indicating differential dysfunctions. The surface expression of programmed cell death 1 ligand 1 (PD-L1) and indolamine 2,3-dioxygenase (IDO) increased while presentation molecule MHC expression decreased. Additionally, increased secretion of TGF-β assists in immune escape from glioblastomas. Tumor-associated macrophages (TAMs) and regulatory T (Treg) cells also facilitate the increase of inhibitory immune checkpoints and secrete TGF-β and IL-10, which downregulate the activation of effector T cells. Exhausted dendritic cell (DC) expression inhibitory immune checkpoints may exaggerate immune resistance in the draining lymph nodes. Furthermore, the presence of infiltration T lymphocytes (TILs), Tregs and CD4^+^ CD56^+^ T cells with production of IL-4 and IL-13, are associated with the induction of GM-CSF secretion by myeloid suppressor cells. Overall, glioblastoma appears to be a highly immunosuppressive tumor. The figure is adapted from reference [15,16,17,24,25,26,27,28,29,30,31,32].

**Figure 2 ijms-23-01311-f002:**
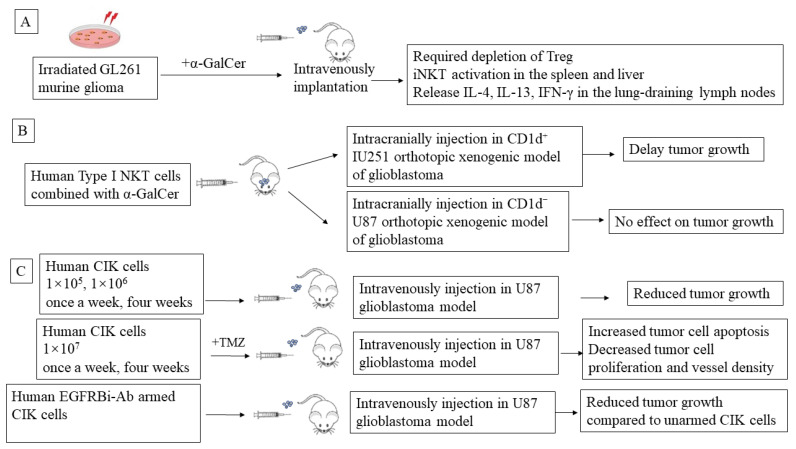
NKT immunotherapy in murine glioblastoma models. (**A**) An irradiated GL261 murine glioma loaded with α-GalCer was implanted intravenously to activate iNKT cells. (**B**) In a CD1d-positive U251, orthotopic xenogenic model of glioblastoma, intracranially co-injected human type I NKT cells with α-GalCer. (**C**) hCIK cells/hEGFRBi-Ab armed CIK were injected into the tail veins of immune-compromised mice bearing U-87MG tumors in their brains. (adapted from references [99,106,107]).

**Figure 3 ijms-23-01311-f003:**
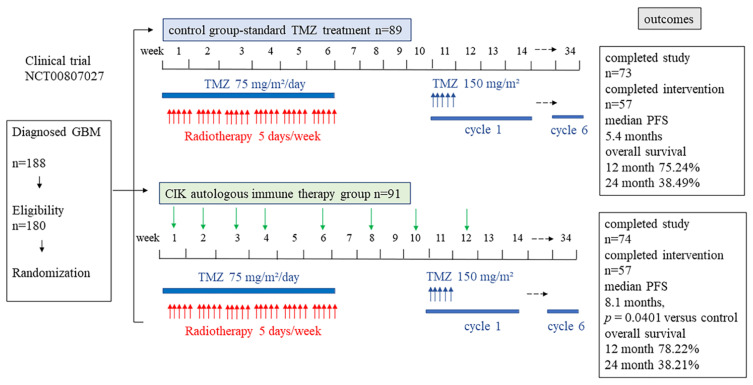
A schematic picture shows the clinical trial NCT00807027, including the study design, randomization, and outcomes.

**Table 1 ijms-23-01311-t001:** List of NKT cells’ classification and properties.

	Type I NKT Cells	Type II NKT Cells	NKT-like Cells
Other names	Invariant NKT(iNKT), Classical NKT cells	Non-classical NKT cells	CD1d-independent NKT cells
CD1d dependent	Yes [60,61,62]	Yes [60]	Unclear [60]
a-GalCer reactive	Yes [60,61,62]	No, but recognize α-GlcADAG [63]	No [60]
TCR α-chain	Vα14-Jα18 (mice) [60,61,62] Vα24-Jα18 (humans) [63,64,65,66]	Diverse [60]	Diverse [60]
TCR β-chain	Vβ8.2, Vβ7 and Vβ2 (mice) [60,61,62] Vβ11 (humans) [47,61,62,63]	Diverse [60]	Diverse [60]
Recognition antigens	α-GalCer [60,61,62]	Sulphatide [67] β-GlcCer, PG, PG, LPC, LPE [68,69,70,71,72,73,74,75,76,77]	MICA/B
NK associated receptors	Mice NK1.1 (human CD161^+^) (resting mature) Mice NK1.1 (human CD161^−^)/low (immature or post-activation) [60,78]	Mice NK1.1 (human CD161^+/−^) [60]	Mice NK1.1 (human CD161^+^) Activation receptors (NKG2C, NKG2D, NKp30, NKp44, NKp46) Inhibitory receptors (CD158a, CD158b, KIR3DL1, and NKG2A) [79,80]
Subsets	CD4^+^ and DN (mice) CD4^+^, CD8^+^ and DN (humans) [60,78]	CD4^+^ and DN (mice) [60]	CD4^+^, CD8^+^ and DN [60]
Cytokines	TH1-like IFN-γ, TNF-α [81] TH2-like IL-4, IL-13 [82] TH17-like IL-17, IL-21, IL-22 [83,84] Treg-like IL-10 [85,86,87] TFH-like-IL-21 [88,89,90]	TH1-like IFN-γ, TNF-α [91] TH2-like IL-4, IL-13 [92,93] TH17-like IL-17, IL-21, IL-22 (mice) [94]	TH1-like IFN-γ Th2-like IL-4, IL-13 [95]

Abbreviations: Type I NKT cells, type I natural killer T cells; iNKT, invariant NKT; TCR, T cell receptor; TH, helper T; Treg, regulatory T; TFH, follicular helper T; DN, double negative.

## Data Availability

Not applicable.

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
