# Peer review of "Recent Development in NKT-Based Immunotherapy of Glioblastoma: From Bench to Bedside"

_ijms, 2022, doi:10.3390/ijms23031311_

Round 1

Reviewer 1 Report

The article presented for review concerns the role of NKT or NKT-like cells in GBM immunotherapy. Although the topic is interesting, the manner of giving information raises several doubts.

Information is often presented chaotically (sentences taken out of context, e.g. line 123, no thought sequence, e.g. lines 29-33, 265-271, sentences which are a list of various factors involved in the process, and their role is not explained in any way, e.g. lines 131-144) that discourages further reading.

The introduction seems to be utterly inconsistent with the topic of the review. It is a collection of little related information on epigenetic changes, GSCs, ways of blocking CBF1 and Wnt pathways in GBM, molecular monitoring of GBM, preclinical models, and all in 15 lines of text.

Analyzing the further content of the article, it is hard not to notice that the article is very similar in its structure to another review published by Emily E. S. Brettschneider and Masaki Terabe in 2021 (10.3390 / cells10071641). The information presented in the chapter "Landscape of unique immune suppression in glioblastoma" and "Natural killer T (NKT) cells" seem to be a summary of the information presented in the article mentioned above. Moreover, the presentation of information in this chapter also leaves much to be desired. From the description of the tumor microenvironment in GBM, it is difficult to conclude what processes occur there and the place/role of NKT cells in such an environment.

Table 1 shows the classification of NKT cells, in which the NK1.1 antigen is listed as a general antigen used for NKT characteristics, while NK1.1. is an antigen characteristic for murine cells. Moreover, the authors use the abbreviation DN in the description of the subtypes. Does that mean double negative? If so, what antigens are the authors referring to? This should be explained below the table.

Could the author explain what he meant by writing "increased levels of transcripts of CD4 + CD25 high FoxP3 + regulatory T (Treg) cells"

or about "α-GalCer-pulsed patient glioblastoma stem-like cells."

Contrary to Brettschneider and Terabe, the reviewed article was supplemented with information about CIK cells. The table presenting the classification of NKT was augmented with a column describing NKT-like cells to which CIK can be classified. The information in this chapter has been described correctly, but it is too little to qualify the article for substantial revision.

Author Response

We would like to thank the editor and reviewers for their valuable comments. We have now revised our manuscript and here providing a point-by-point response. All these changes are highlighted in red in our revised version. We sincerely hope that you will find the revised version suitable for publication in your journal.

Comments from reviewer 1:

Comment 1: The article presented for review concerns the role of NKT or NKT-like cells in GBM immunotherapy. Although the topic is interesting, the manner of giving information raises several doubts. Information is often presented chaotically (sentences taken out of context, e.g. line 123, no thought sequence, e.g. lines 29-33, 265-271, sentences which are a list of various factors involved in the process, and their role is not explained in any way, e.g. lines 131-144) that discourages further reading.

Response: Thank you for your comment. We have now rephrased the concerned text as follows:

Line 29-33: In addition, the high relapse rate, unfavorable prognosis, and post-treatment symptoms in GBM pose a serious clinical challenge. Of note, patients usually develop GBM sporadically, however, a study described 10 inherited variants (known as glioma risk variants) near 8 genes that may influence the risk of developing glioma [12]].

Line 123: To extend the use of immune checkpoint inhibitors in GBM, a study revealed that tumor-infiltrating CD8+ T cells were exhausted in patients with newly diagnosed GBM and PD-1 blockade could revive the CD8+ TIL response [28].

Line 265-271: Given the ability of NK cells to efficiently kill malignant cells and their utility in clinical practice (albeit still moderate), continuous efforts are underway to improve the efficacy of NK cell-based therapies. In particular, type I NKT cell-targeted adaptive therapies, such as autologous α-GalCer-pulsed antigen-presenting cells hold great promise as options for cancer treatment. In this context, a study investigated the impact of a vaccine composed of irradiated autologous tumor cells pulsed with α-GalCer, an adjuvant that stimulates iNKT cells, as an effective treatment against tumors in a murine glioma model [117].

In addition, we have edited the complete paragraph including lines 131-144 in our revised version.

Comment 2: The introduction seems to be utterly inconsistent with the topic of the review. It is a collection of little related information on epigenetic changes, GSCs, ways of blocking CBF1 and Wnt pathways in GBM, molecular monitoring of GBM, preclinical models, and all in 15 lines of text.

Response: Thank you, as you have suggested, we have now provided more information in the introduction section, as follows: 

GBM ranks high in cancer epigenomics due to the clinical relevance of mutations in isocitrate dehydrogenase-1 (IDH1) and abnormal methylation of the O6 methylguanine DNA methyltransferase (MGMT) gene. The routine analysis of the methylation status of the promoter region of the MGMT is beneficial for an effective response to temozolomide-based chemotherapy [3]. Interestingly, several microRNAs (miRNAs) have been identified in GBM and are thought to play important roles in initiation, progression and response to therapy (PMID: 34434499, 4, 5).Of interest, these miRNAs are often combined with MGMT status to assign patients to high and low risk groups (PMID: 27302927, PMID: 29695934).

Another layer of complexity in GBM relates to microanatomical compartments representing specialized tumor niches within the tumor microenvironment that regulate metabolic needs, immune surveillance, survival, invasion, as well as maintenance of cancer stem cells (PMID: 27088132). Certainly, the contribution of complex microenvironments surrounding glioblastoma tumors, which primarily is composed of cell-produced soluble factors, extracellular matrix components, resident cells (e.g., astrocytes, endothelial cells, pericytes, and microglia) and recruited cells (e.g., bone marrow-derived macrophages, tumor-infiltrating lymphocytes (TILs) cannot be excluded [12]. To mention, there have been considerable progress concerning therapeutically targeting of the microenvironment in GBM (PMID: 28292436), however, the clinical implementation is pending.

Comment 3: Analyzing the further content of the article, it is hard not to notice that the article is very similar in its structure to another review published by Emily E. S. Brettschneider and Masaki Terabe in 2021 (10.3390 / cells10071641). The information presented in the chapter "Landscape of unique immune suppression in glioblastoma" and "Natural killer T (NKT) cells" seem to be a summary of the information presented in the article mentioned above. Moreover, the presentation of information in this chapter also leaves much to be desired. From the description of the tumor microenvironment in GBM, it is difficult to conclude what processes occur there and the place/role of NKT cells in such an environment.

Response: Thank you for your comment. We are aware of the article you mentioned, since both are review articles, a slight overlap of some sections can be expected due to the use of similar references. However, we emphasize not only the role of NK cells in a specific suppressive environment, but also discuss other issues related to glioblastoma that are distantly dependent on NK cells. To get further uniqueness, we have now added about glioblastoma and NKG2D aspect, which is an active receptor on the NK cell as follows:

Natural killer group 2-member D (NKG2D) is known to induce cytotoxicity and cytokine production by NKT cells upon binding to its ligands. It is also well established that NKG2D recognize number of legends [MHC class I polypeptide-related sequence A (MICA), MHC class I polypeptide-related sequence B (MICB), and UL16-binding proteins 1–6 (ULBP1-6)]. Of interest, these ligands are also expressed on human glioma cells, in vitro (PMID: 14695218), in vivo (PMID: 16891318) and on glioma stem cells (PMID: 24327582, PMID: 29356965). Recently, one study not only confirmed the high expression of NKG2DL in human glioma cell lines, cancer stem cells and tumor samples, but also suggested that CAR-T cells expressing NKG2D are an encouraging therapeutic approach for glioma patients (PMID: 31288857). An interesting study described the potential mechanism of immune escape in which glioblastoma cells produce a soluble protein LDH5 that induces the expression of NKG2D ligands on the surface of healthy myeloid cells (PMID: 25136121). Of note, combining NKG2D-based immunotherapies with temozolomide (TMZ) or irradiation (IR) has also been suggested (PMID: 29162646).

Please also see our response to comment 10 /Reviewer 4.

Comment 4: Table 1 shows the classification of NKT cells, in which the NK1.1 antigen is listed as a general antigen used for NKT characteristics, while NK1.1. is an antigen characteristic for murine cells. Moreover, the authors use the abbreviation DN in the description of the subtypes. Does that mean double negative? If so, what antigens are the authors referring to? This should be explained below the table.

Response:  We apologies for this text error. We have now corrected this in the table and also added the abbreviation for DN (as double negative) in the revised version.

Comment 5: Could the author explain what he meant by writing "increased levels of transcripts of CD4 + CD25 high FoxP3 + regulatory T (Treg) cells" or about "α-GalCer-pulsed patient glioblastoma stem-like cells."

Response: Thank you, we now rephrase the first sentences as:

In addition to some barriers to mount an effective immune response against GBM, one study performed T cell mRNA expression profiles in GBM patients and found that genes associated with T cell activation were significantly reduced in CD4+ and CD8+ T cells, while expression of inhibitory genes was increased in the immunosuppressive Treg subset (PMID: 17189402).

Regarding your concerns about "α-GalCer-pulsed patient glioblastoma stem-like cells", in that particular study, the patient's glioblastoma cells were cultured in DMEM/F-12 medium containing a range of growth factors to generate glioblastoma stem-like cells as neurospheres. Also, the expression level of CD1d on the patient's glioblastoma-derived stem cells was found to be lower than that on the patient's original glioblastoma cells. We have now revised this information and also included the abbreviation for alpha-GalCer (alpha-galactosylceramide).

Comment 6: Contrary to Brettschneider and Terabe, the reviewed article was supplemented with information about CIK cells. The table presenting the classification of NKT was augmented with a column describing NKT-like cells to which CIK can be classified. The information in this chapter has been described correctly, but it is too little to qualify the article for substantial revision.

Response: Thank you for your insightful comment. Strikingly, there is very limited information on CIK in the context of Glioblastoma. There are certain factors that we have suggested to the oncology community through our recent article: 30 years of CIK cell therapy: recapitulating the key breakthroughs and future perspective (PMID: 34886895, J Exp Clin Cancer Res 2021) that will help to further optimize clinical trials, including Glioblastomas. We in our capacity are currently conducting experiments to confirm the efficacy of combined PD-1/PD-L1 inhibitors with CIK cells against GBM cell lines, but those analyses will be reported in an independent manuscript.

Reviewer 2 Report

The paper is well written and provides clear information about the biological characteristics and clinical potentialities of NKT cells in GBM.

I have only minor comments regarding the figure that in some cases are hard to read. I suggest overwriting the original pictures increasing when possible the font size.

In addition, I would suggest checking the references. I understand that citing the milestone of the proposed research can include publication of many years ago, but some of them seem a little bit outdated.

Author Response

We would like to thank the editor and reviewers for their valuable comments. We have now revised our manuscript and here providing a point-by-point response. All these changes are highlighted in red in our revised version. We sincerely hope that you will find the revised version suitable for publication in your journal.

Comment 1: The paper is well written and provides clear information about the biological characteristics and clinical potentialities of NKT cells in GBM. I have only minor comments regarding the figure that in some cases are hard to read. I suggest overwriting the original pictures increasing when possible the font size.

Response: Thank you for supporting our manuscript. As you suggested, we have increased the font size in the original figures.

Comment 2: In addition, I would suggest checking the references. I understand that citing the milestone of the proposed research can include publication of many years ago, but some of them seem a little bit outdated.

Response: Thank you, we updated the references.

Reviewer 3 Report

thanks

Author Response

We would like to thank the editor and reviewers for their valuable comments. We have now revised our manuscript and here providing a point-by-point response. All these changes are highlighted in red in our revised version. We sincerely hope that you will find the revised version suitable for publication in your journal.

Comment 1: English language and style are fine/minor spell check required

Response: Thank you for supporting our manuscript. We have checked this thoroughly in our revised version.

Reviewer 4 Report

The manuscript entitled „Recent Development in NKT-Based Immunotherapy of Glioblastoma: From Bench to Bedside” by Li et al. aims for a review of recent developments in NKT based immunotherapy of glioblastoma. While the topic is in general of high interest, some parts of the review are defocused.

A review article should summarize the current state of understanding on a topic and present an unbiased summary of the current understanding of the topic. In my opinion, this manuscript does fulfil these criteria, however it is partly defocused. While a review article contains a large amount of detailed information, its structure and flow are also important, which should be improved for the supplied manuscript.

I recommend major revision.

Some points of criticism:

-In abstract authors wrote that there is “lack of tumor infiltration lymphocytes (TILs)”. This statement is not support by actual publications

-The introduction should be rewritten as it contains facts not set into context of the review and many abbreviations are not explained e.g. GLS, CBF1, EMT line 44-51

-2.1: BBTB composition is not explained.

-2.3 Glioblastoma tumor microenvironment: the focus of the authors is on immunotherapy, still in the chapter 2.3 different aspects are missing, astrocytes, tumor associated macrophages etc. The components of microenvironment determinate the progression and therapeutic answer.  Given the title of this chapter, these aspects should at least briefly be mentioned and this part should be rewritten und restructured.

Line 145: The statement about origin of microglia is not correct, since endogenous microglial population evolves during early development from yolk sac.

3: line 203-209 UC and intestine are not relevant to chosen topic in the way presented here.

-6.3: 3 of 4 clinical studies were submitted many years ago and are still not recruiting. This is a strong limitation. This aspects need to be mentioned in the manuscript. Only one clinical study was finished:

NCT01235845 is listed in 2010 and is not yet recruiting

NCT02496988 is listed in 2015 and is not yet recruiting

NCT02494804 is listed in 2015 and not yet recruiting

To this reviewers opinion the authors over-interpreted the results of this one single study, since the CIK immunotherapy group did not show evidence of a beneficial effect on overall survival.

-7: Conclusions and future perspective should be rewritten. Line 358-385 should be moved to the chapter about preclinical model or other extra chapter.

-Figure 3: C is confusing. Please revise it.

The authors may consider inserting more tables for better traceability.

Author Response

We would like to thank the editor and reviewers for their valuable comments. We have now revised our manuscript and here providing a point-by-point response. All these changes are highlighted in red in our revised version. We sincerely hope that you will find the revised version suitable for publication in your journal.

Comments from reviewer 4:

The manuscript entitled „Recent Development in NKT-Based Immunotherapy of Glioblastoma: From Bench to Bedside” by Li et al. aims for a review of recent developments in NKT based immunotherapy of glioblastoma. While the topic is in general of high interest, some parts of the review are defocused. A review article should summarize the current state of understanding on a topic and present an unbiased summary of the current understanding of the topic. In my opinion, this manuscript does fulfil these criteria, however it is partly defocused. While a review article contains a large amount of detailed information, its structure and flow are also important, which should be improved for the supplied manuscript. I recommend major revision. Some points of criticism:

Comment 1: In abstract authors wrote that there is “lack of tumor infiltration lymphocytes (TILs)”. This statement is not support by actual publications

Response: Thank you for your comment. We have now shifted this text to later section of the manuscript and explained it with more references.

Comment 2: The introduction should be rewritten as it contains facts not set into context of the review and many abbreviations are not explained e.g. GLS, CBF1, EMT line 44-51

Response: Thank you. As per your suggestions, we have now revised it.

Comment 3: BBTB composition is not explained.

Response: Thank you, we have now enhanced the information as follows:

The scenario is much more complex for the blood-brain tumor barrier (BBTB), which reflects the complex regional heterogeneity of immune cell populations compared to the BBB [24]. The BBTB is generally considered ‘leakier’ than the BBB, which is characterized by aberrant pericyte distribution and loss of astrocytic endfeet and neuronal connections [25, 26]. It has been confirmed that the BBB integrity is disrupted by invading of glioma cells [27, 28].

Comment 4:  Glioblastoma tumor microenvironment: the focus of the authors is on immunotherapy, still in chapter 2.3 different aspects are missing, astrocytes, tumor associated macrophages etc. The components of microenvironment determinate the progression and therapeutic answer.  Given the title of this chapter, these aspects should at least briefly be mentioned and this part should be rewritten und restructured.

Response: Thank you for your comment. We have now included more information, as follows:

Given that TME in GBM provides an unfavorable niche for the function of NK cells, i.e., understanding the orchestration of TME constituents is also crucial. In particular, it is important to mention the interaction of microglia and astrocytes (glioma-associated microglia/macrophages: GAMs) in GBM (PMID: 30123112, PMID: 29389898), as it has already been shown that targeting these cells can have a substantial impact on GBM. One study suggested that the immunomodulatory properties of NK cells, mainly their ability to secrete pro-inflammatory cytokines and to influence microglia and macrophage activity, could be used to enhance the efficacy of passive immunotherapy targeting tumor-associated antigens in GBM (PMID: 24575382). Similarly, one study reported that GAMs, stimulated by brain-released BDNF, increased NK cell infiltration and activation thus contributing to the modulating glioma expansion (PMID: 29286001). An interesting study showed that CCP treatment caused the GBM tumor to acquire M1-type macrophages and activated NK cells ((PMID: 30041669). Authors further suggested that M1 microglia-derived MCP-1 may recruits both cell types (M1 macrophages and activated NK cells) into the GBM tumor.

Comment 5: Line 145: The statement about origin of microglia is not correct, since endogenous microglial population evolves during early development from yolk sac.

Response: We apologize, our context was mispronounced. We have now expanded the information about microglia as we respond to your comment 4 above.

Comment 6: line 203-209 UC and intestine are not relevant to chosen topic in the way presented here.

Response: Thank you, we have now removed this text.

Comment 7: 3 of 4 clinical studies were submitted many years ago and are still not recruiting. This is a strong limitation. This aspects need to be mentioned in the manuscript. Only one clinical study was finished. NCT01235845 is listed in 2010 and is not yet recruiting, NCT02496988 is listed in 2015 and is not yet recruiting, NCT02494804 is listed in 2015 and not yet recruiting

Response: We have now specifically mentioned the current status of clinical trials in the revised version.

Comment 9: To this reviewers opinion the authors over-interpreted the results of this one single study, since the CIK immunotherapy group did not show evidence of a beneficial effect on overall survival.

Response: Thank you. That is correct, lack of additional clinical trials has led to this conclusion. Strikingly, there is very limited information on CIK in the context of Glioblastoma. There are certain factors that we have suggested to the oncology community through our recent article: 30 years of CIK cell therapy: recapitulating the key breakthroughs and future perspective (PMID: 34886895, J Exp Clin Cancer Res 2021) that will help to further optimize clinical trials, including Glioblastomas. We in our capacity are currently conducting experiments to confirm the efficacy of combined PD-1/PD-L1 inhibitors with CIK cells against GBM cell lines, but those analyses will be reported in an independent manuscript.

Comment 10: Conclusions and future perspective should be rewritten. Line 358-385 should be moved to the chapter about preclinical model or other extra chapter.

Response: Thank you. As you suggested we have shifted the line and included some more information about conclusions and future perspective, as follows:

This review highlights the role of NKT cells in glioma, with particular emphasis on their subtypes. Based on the current status of NK cells and their efficacy in past studies, it is reasonable to speculate that if we enhance their function, we can expect potentially potent anti-tumor immunity. However, this would require a better understanding of how TME components are orchestrated, as they appear to be the main culprits in providing an unfavorable niche for NK cell function. Since NK cell activation requires the presence of proinflammatory cytokines and engagement of surface receptors, the degree to which patients' own genetic setup contributes to the variable response of these particular cells to therapy, also requires reconsideration in preclinical trials. This probable factor may also account for the possible escape of GBM cells (due to their heterogeneous profile) from immune surveillance by exhibiting resistance to NK cells toxicity. Additionally, to overcome the limited ability of NK cells, a combinatorial NK/CIK cell approach (due to their biological similarities) can be considered for GBM treatment. Specifically, the possible efforts on α-GalCer-loaded DC-CIK cells can be considered at least partially successful if iNKT cell levels can be restored in GBM patients. Overall, encouraging the ongoing discussion about individualized therapeutic approaches in GBM, and cancer in general, may help to limit the shortcomings of future clinical trials.

Comment 11: Figure 3: C is confusing. Please revise it.

Response: Thank you. We have now revised it.

In conclusion, we closely followed the recommendations of the reviewers. We feel that our manuscript has been significantly improved and hope that you will find it now suitable for publication in your journal.

Thank you very much.

Yutao Li (on behalf of all co-authors)

Round 2

Reviewer 1 Report

Although the authors significantly revised the manuscript, it still needs further refinement. It concerns mainly the Conclusion and future perspectives section. It is unclear why the authors write here about NK cells, while NKT cells are the main subject of the review.  It is not the same. Moreover, the information presented in the section doesn't fit the general part of the manuscript. In this place, the different perspectives of the use and the development of NKT/CIK-based therapeutic strategy should be discussed. Asking the questions that should be resolved in the further study would also enhance the discussion's quality.
Additionally, the content of the Introduction still doesn't fit the rest of the article. Why do the authors concentrate on epigenetic changes in GBM, although all these data are not significant in data concerning the NKT in GBM?  If yes, the connection should be clearly explained.
In line 334, the authors write, "To overcome the disadvantages of suppressive tumor environment, PD-1/PD-L1 blockade might be a valuable option to enhance the cytotoxicity of iNKT cells [116] or CIK cells in the future". The information needs a further extension.

Author Response

We would like to thank the editor and reviewers for acknowledging our work. Since all the other three reviewers (Reviewer 2, 3 and 4) are satisfied with the previous revision, herein, we are providing our point-by-point response to the valuable comments from Reviewer 1. All these changes are highlighted in red in our revised version. We sincerely hope that you will find the revised version suitable for publication in your journal.

Comments from reviewer 1:

Comment 1: Although the authors significantly revised the manuscript, it still needs further refinement. It concerns mainly the Conclusion and future perspectives section. It is unclear why the authors write here about NK cells, while NKT cells are the main subject of the review.  It is not the same. Moreover, the information presented in the section doesn't fit the general part of the manuscript. In this place, the different perspectives of the use and the development of NKT/CIK-based therapeutic strategy should be discussed. Asking the questions that should be resolved in the further study would also enhance the discussion's quality.

Response: Thank you, we have now rephrased the conclusion and future perspectives section as below:

This review highlights the role of NKT cells in glioma, with particular emphasis on their subtypes. Based on the efficacy of NKT cells in previous studies, it is reasonable to speculate that there is an urgent need to enhance their function, primarily by regulating their activation throughout the tumor progression, in order to maintain their anti-tumor functions. This, in turn, may help to balance the activity of NKT cells within the intracellular signaling cascade, thereby reducing their chances of immune escape. Additionally, we need a better understanding about the orchestration of TME components, as they appear to be the main culprits in providing an unfavorable niche for the function of immune cells, including NKT cells. Considering the biological similarities between NKT and CIK cells, a combinatorial approach can be considered to enhance immune surveillance. In this respect, the simultaneous expansion and possible activation of these NKT/CIK cells can be established and tested in preclinical models. The preliminary results from the combination of iNKT-targeted dendritic cell (DC)  vaccine appear to be encouraging in GBMs, and further consideration with α-GalCer-loaded DC-CIK cells may also help to expand this avenue. Clearly, more research is required to elucidate the therapeutic potential of NKT-based approaches. Crucially, the extent to which patients' own genetic settings contribute to variable responses to therapies based on these cells also needs to be reconsidered.

Comment 2: Additionally, the content of the Introduction still doesn't fit the rest of the article. Why do the authors concentrate on epigenetic changes in GBM, although all these data are not significant in data concerning the NKT in GBM?  If yes, the connection should be clearly explained.

Response: Thank you, we merely wanted to address the clinical prognostic possibilities in GBM, therefore we have now rephrased this particular sentence as below.

GBM ranks high on the cancer spectrum owing to epigenetic-based prognostic markers, e.g. IDH1 gene mutations with high global DNA methylation levels and hypermethylation in the MGMT gene promoter region play a pivotal role to make clinical decisions.

Comment 3: In line 334, the authors write, "To overcome the disadvantages of suppressive tumor environment, PD-1/PD-L1 blockade might be a valuable option to enhance the cytotoxicity of iNKT cells [116] or CIK cells in the future". The information needs a further extension.

Response: Thank you for your comment. We have now rephrased this sentence as below.

To overcome the disadvantages of suppressive tumor environment, PD-1/PD-L1 blockade might be a valuable option to enhance the cytotoxicity of iNKT cells. In this context, a study showed that the co-administration of anti-PDL1 antibody and alpha-galactosylceramide (αGalCer)-pulsed APCs enhances iNKT cell-mediated antitumor immunity [109]. Similarly, a critical role of PD-1/PD-L costimulatory pathway in the alpha GalCer-mediated induction of iNKT cell anergy as a possible target for immunotherapies has been discussed [110]. Since PD-L1 expression is a predictive biomarker for CIK cell-based immunotherapy and PD-1 blockade has been shown to enhance CIK cell cytotoxicity [111], PD-Ls/NKT/CIK axis needs consideration.

In conclusion, we closely followed the recommendations of the reviewers. We feel that our manuscript has been significantly improved and hope that you will find it now suitable for publication in your journal.

Thank you very much.

Amit Sharma (on behalf of all co-authors)

Round 3

Reviewer 1 Report

The manuscript has been completed according to the reviewer's comments.